# Cost-effectiveness of providing university students with a mindfulness-based intervention to reduce psychological distress: economic evaluation of a pragmatic randomised controlled trial

Adam P Wagner [ORCID],[1,2] Julieta Galante [ORCID],[3,4] Géraldine Dufour,[5,6] Garry Barton,[2] Jan Stochl,[3,7] Maris Vainre [ORCID],[8] Peter B Jones [ORCID] [1,3]

**Correspondence to**
Dr Adam P Wagner;
Adam.Wagner@uea.ac.uk

## ABSTRACT

**Objective** Increasing numbers of young people attending university has raised concerns about the capacity of student mental health services to support them. We conducted a randomised controlled trial (RCT) to explore whether provision of an 8 week mindfulness course adapted for university students (Mindfulness Skills for Students—MSS), compared with university mental health support as usual (SAU), reduced psychological distress during the examination period. Here, we conduct an economic evaluation of MSS+SAU compared with SAU.

**Design and setting** Economic evaluation conducted alongside a pragmatic, parallel, single-blinded RCT comparing provision of MSS+SAU to SAU.

**Participants** 616 university students randomised.

**Primary and secondary outcome measures** The primary economic evaluation assessed the cost per quality-adjusted life year (QALY) gained from the perspective of the university counselling service. Costs relate to staff time required to deliver counselling service offerings. QALYs were derived from the Clinical Outcomes in Routine Evaluation Dimension 6 Dimension (CORE-6D) preference based tool, which uses responses to six items of the Clinical Outcomes in Routine Evaluation Outcome Measure (CORE-OM; primary clinical outcome measure). Primary follow-up duration was 5 and 7 months for the two recruitment cohorts.

**Results** It was estimated to cost £1584 (2022 prices) to deliver an MSS course to 30 students, £52.82 per student. Both costs (adjusted mean difference: £48, 95% CI £40–£56) and QALYs (adjusted mean difference: 0.014, 95% CI 0.008 to 0.021) were significantly higher in the MSS arm compared with SAU. The incremental cost-effectiveness ratio (ICER) was £3355, with a very high (99.99%) probability of being cost-effective at a willingness-to-pay threshold of £20 000 per QALY.

**Conclusions** MSS leads to significantly improved outcomes at a moderate additional cost. The ICER of £3355 per QALY suggests that MSS is cost-effective when compared with the UK's National Institute for Health and Care Excellence thresholds of £20 000 per QALY.

---

### STRENGTHS AND LIMITATIONS OF THIS STUDY

⇒ Contrasting with many evaluations of mindfulness, a key strength of this study/economic evaluation is that it drew on data from a large (n>600) randomised controlled trial that followed up participants for up to 12 months.

⇒ A key limitation is the narrow costing perspective adopted—the university counselling service—that does not capture all costs of mental health support and will miss wider impacts.

⇒ Additionally, we have focused exclusively on wage costs (including on-costs)—we have not considered wider overheads or other resources (such as room hire), subsequently underestimating costs.

⇒ We have not considered the wider questions about whether Mindfulness Skills for Students is a cost-effective alternative to other innovations, or whether it would be an affordable option to offer to a much larger proportion of students.

**Trial registration number** Australian and New Zealand Clinical Trials Registry, ACTRN12615001160527.

## BACKGROUND

Depression is among the top causes of morbidity, generating a huge burden on populations all over the world.[1] This trend is also seen in youth, with the prevalence of a probable mental health disorder among those aged 17–19 in England reaching almost one in four in 2022.[2] University counselling services in the UK have seen a rising trend in the proportion of students asking for mental health support[3]; indeed, it has been noted that the increase in student numbers seeking counselling is greater than the growth of overall student numbers.[4] Similar issues have been noted in the university sector beyond

England.[5 6] Thus, an effective preventative intervention is needed to address this growing demand and need.[7]

A widely used working definition of mindfulness is 'the awareness that emerges through paying attention on purpose, in the present moment, and non-judgmentally to the unfolding of experience moment by moment'.[8] It is popular among students,[9] and some of the popularity may be because it seeks to teach skills, rather than directly addressing mental health issues—it is non-stigmatising.[10]

Given the growing demand for counselling services and limited university funding, it is important to have evidence for the effectiveness and cost implications of associated interventions.[11] Thus, when the University of Cambridge's (UoC's) University Counselling Service (UCS) developed the 'Mindfulness Skills for Students' (MSS) programme, they evaluated it in the Mindful Student Study: a large randomised controlled trial (RCT).[12] It has been demonstrated that, compared with support as usual (SAU), MSS+SAU reduces students' psychological distress during the academic examination period, 3–6 months after randomisation.[13] MSS participants were followed up for 12 months postrandomisation: compared with SAU, MSS+SAU continued to have small but significant reductions in distress at this time.[14] This result suggests that the MSS programme, via reducing psychological distress, could prevent future cases of mental ill-health.[15] Here, we conduct an economic evaluation of MSS to explore its costs.

## METHOD
### Participants
The Mindful Student Study was a pragmatic randomised trial at the UoC in the UK.

The trial inclusion criteria were: (a) undergraduate or postgraduate UoC students and (b) who were interested in attending at least seven sessions of the course. The exclusion criteria were: (a) severe anxiety or depression at that time; (b) severe mental illness such as psychosis; (c) recent bereavement and (d) any other serious health problem that would affect their ability to engage with the course. The selection criteria were assessed by the students themselves.

The study was advertised widely to the student community, using physical (posters), social media (eg, Facebook and Twitter) and information sessions—see 12 for further detail. Where students agreed to take part, they were emailed a personal link to an online baseline questionnaire. On completing this questionnaire, participants underwent 1:1 randomisation to receive either: an 8 week mindfulness course adapted for university students plus support as usual (MSS+SAU); or SAU alone. SAU consisted of the possibility of accessing, if the student desired, comprehensive UCS support in addition to other health support available from the UoC and its colleges, and from health services including the National Health Service (NHS), external to the UoC. Participants randomised to SAU were guaranteed a space in the

following year's mindfulness courses and were requested to inform the team if they decided to learn mindfulness elsewhere during the follow-up period. Further detail on the trial can be found in 12 and 13.

Students were recruited in two cohorts: at the beginning of the autumn (Michaelmas) term and at the beginning of the spring (Lent) term. Both cohorts were followed up at: point of recruitment (T0); after the delivery of the mindfulness courses (SAU having *not* received the course) (T1); during the examination term (T2) and 12 months after recruitment (T3). The date of examination term follow-up, T2, was common across cohorts, but the date of other follow-up points differed by cohort: the relationships are depicted in figure 1. Thus, the follow-up duration at T2 was approximately 7 months for the Michaelmas cohort and 5 months for the later recruited Lent cohort.

This study was approved by the Cambridge Psychology Research Ethics Committee on 25 August 2015 (reference: PRE.2015.060). The study is registered with the Australia and New Zealand Clinical Trials Registry (ACTRN12615001160527). The Health Economic Analysis Plan (HEAP) that this economic evaluation follows was prespecified and is available online.[16]

### Costs
Costs were estimated from the perspective of the UCS' operating budget (primarily covering salaries of staff; excludes room hire)—hereafter referred to as the 'UCS perspective'. Use of SAU UCS offerings were extracted from the UCS record system (consequently, there is no missing data on UCS service use) following participant agreements and strict confidentiality of access protocols.

At the time of this evaluation, routine UCS services were categorised into three broad groups: 'individual' sessions providing support to one student (includes assessment sessions); 'workshops'—one-off offerings that more than one student attends—and 'groups'—a number of sessions attended by multiple students (where a student stops attending a group, their place cannot be used by another student). In consultation with the then UCS head (GD), who consulted the UCS senior management team, we determined the usual staff grade delivering each offering and the associated typical time taken to prepare, deliver and record (eg, completing notes) each of these session types. To calculate the cost to the UCS of a given offering, we multiplied the time taken to prepare, deliver and record it by the hourly rate of employing staff at the required grade. For workshops and groups, the cost per student (unit cost) of an offering is determined by dividing its total cost by the number of students typically attending each offering (details given in the online supplemental file).

The total cost of delivering a course of mindfulness (the intervention) to 30 students (maximum capacity of a course) was determined through consultation with the then UCS head (GD) and the course teacher. Accordingly, the activities involved in delivering a course were

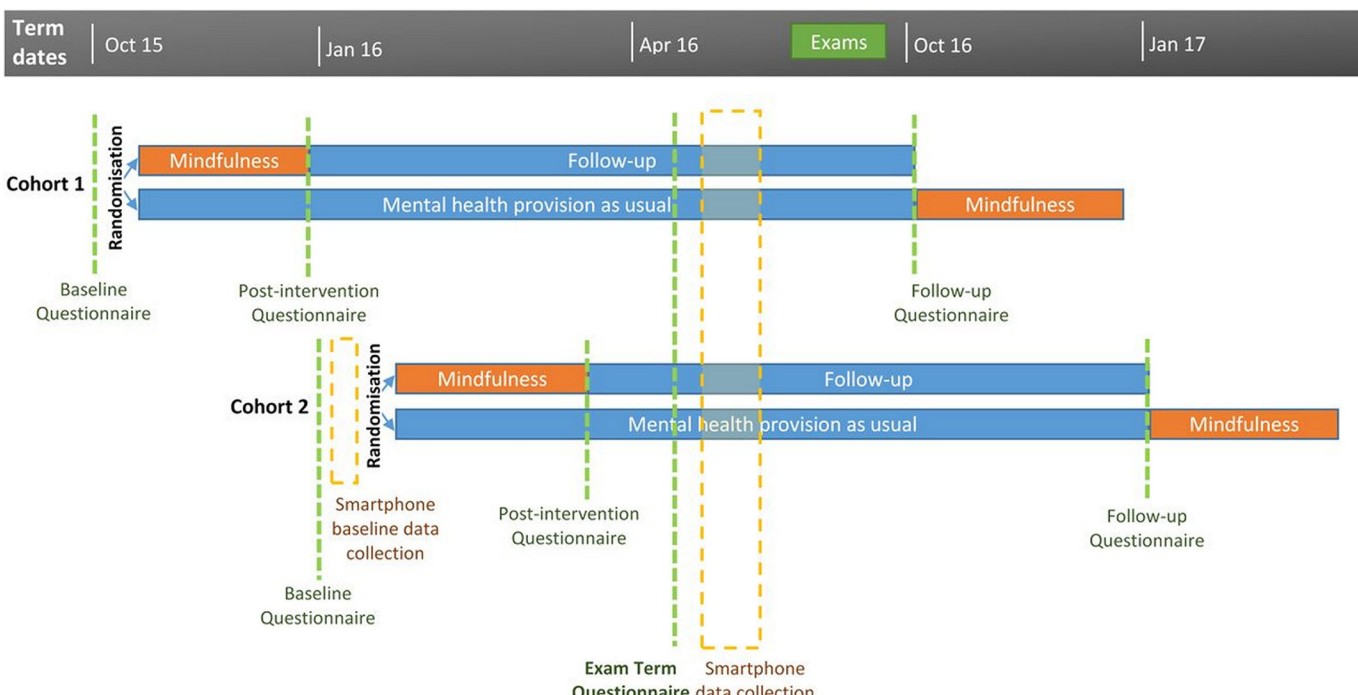

**Figure 1** Participant timeline. Reproduced from Galante *et al* (2016).[12]

identified, along with the total time, and grade of staff, delivering each activity. Total times were multiplied by appropriate hourly rates of staff delivering each course activity and summed to give the total cost of delivering a mindfulness course. The cost per student (unit cost) of delivering the course was determined by dividing this total cost by the capacity of each class (n=30 students). This cost was ascribed to each of the students in the MSS+SAU arm (multiple courses were being delivered in parallel for each cohort of MSS+SAU). This cost was irrespective of student attendance: the nature of the mindfulness course is such that a student cannot join part way through, so a non-attending student cannot be 'replaced'; this is in line with other UCS groups.

Hourly cost of employing staff were taken from the UoC Human Resources website and include on-costs (such as pension and national insurance contribution—see online supplemental file) but no further overheads. They were the rates as applied from March 2022, and so can be considered to apply to the 2022/2023 financial year. Costs are reported in pound sterling.

### Outcomes

The primary outcome measure was the participant completed Clinical Outcomes in Routine Evaluation Outcome Measure (CORE-OM), a 34-item generic questionnaire designed to evaluate efficacy and effectiveness across multiple disciplines delivering psychological therapies.[17] The CORE-OM has been found to have strong convergent validity, internal and test-retest reliability and sensitivity to change,[18] and has been extensively used with UK university students.[19] Each CORE-OM item is scored on a five-point scale ranging from 0 (not at all) to 4 (most or all the time). The corresponding sum-total score ranges

0–136; this is usually divided by the number of completed items to form a total mean score. A higher mean score indicates more distress. CORE-OM was collected at each of the data collection points (T0–T3); a comparison of CORE-OM between arms during the examination term (T2) was the primary (clinical effectiveness) outcome.[12 13]

The Clinical Outcomes in Routine Evaluation Dimension 6 Dimension (CORE-6D) is a preference-based tool to derive mental health specific health state utility, based on a subset of six questions/dimensions from the CORE-OM[20]: using time trade-off methods,[21] a sample of the general public were surveyed for their preferences for different health states as described by the CORE-6D.[22] Scores generated from the survey are converted to an index score that can be used as a valuation of all health states represented by the CORE-6D: details of the steps for these calculations are provided in an Appendix to the HEAP.[16] Here, utility scores were calculated for T0–T3 using this valuation. In line with the primary clinical-effectiveness outcome, the base case (primary) economic analysis uses quality adjusted life years (QALYs) from T0 to T2 as outcome; these are calculated via the area under the curve approach[23] with linear interpolation between points. Given different periods between cohorts in which to accrue QALYs (see figure 1: Michaelmas cohort—227 days; Lent—147 days), QALYs are reported separately for each cohort (and subsequent regressions adjust for cohort). Additionally, we also calculated QALYs from T0 to T3.

The Warwick-Edinburgh Mental Well-being Scale (WEMWBS) was used at each time point to measure a broad conception of subjective well-being.[24] The WEMWBS has 14 items, each scored from one ('none of

the time') to five ('all of the time'). A higher total score (ranging from 14 to 70) indicates higher well-being. The WEMWBS has good validity, internal consistency and test-retest reliability in samples of both UK students and general population.[25]

## Analyses

To reduce the risk of bias[26] and improve precision,[27] we sought to address missing data, rather than conducting a complete case analysis (CCA). Following recommendations for within trial analyses of cost-effectiveness, patterns of missing data were explored to understand the potential missing data mechanisms.[27] Resource use information was complete as it came from internal UCS records. Where missingness occurred on the outcome scales, it nearly always meant that *no* items had been completed (see online supplemental file). Thus, we focused on summary measures of the outcomes (eg, utilities, CORE-OM mean-score and WEMWBS total) in further missing data considerations. Details of missingness by outcome at each time point, overall and by arm, are given in the online supplemental file: missingness increased by time point, and was between 5% and 10% higher in the SAU at T1 and T2, but were similar at T3. As advised in [27], we explored associations between missingness and both baseline variables and observed outcomes (see online supplemental file). The associations indicated that the data was not missing completely at random—meaning that CCA would not be appropriate. In line with the analysis devised in [12] and conducted in [13], we assumed the data was missing at random, and used multiple imputation (MI) to create fifty datasets (following the rule-of-thumb from [28] that the number of datasets should equal the percentage of missing data). Note that the level of missing data at T3 (≈45%) is notably higher than at T2 (≈27%), with the latter time point being the most crucial for the base case which draws data from T0 to T2 (the impact of imputing missing data is explored in the sensitivity analyses—see below).

The R package *mice*[29] was used to conduct MI, separately for each arm (as advised in [27]) using the significant (p<0.1) predictors of missingness and those variables included in the final regression models (as advised in [28]): see online supplemental file for details. The separate datasets were combined using Rubin's rules.[30]

Costs and outcomes were analysed simultaneously with bivariate regression models, allowing for correlations between costs and outcomes to be incorporated (in contrast to separate regressions). Such regressions are generally robust to skewed data.[31] Data was analysed using an intention-to-treat approach: participants were analysed in the group to which they were randomised. All regressions (both cost and outcome) included covariates for arm, sex and cohort. Outcome regressions additionally included the baseline measure of the corresponding outcome (or utility, when analysing QALYs).

The estimated coefficient of arm in these regressions allowed the incremental cost-effectiveness ratio (ICER)

to be estimated[32]: in this context, this corresponded to the mean incremental cost divided by the mean incremental effect (for the QALYs, COME-OM and WEMWBS outcomes) for MSS+SAU compared with SAU. However, were either alternative—MSS+SAU or SAU—both less costly and more effective, that alternative would be categorised as 'dominant' and the ICER would not be calculated. Otherwise, the ICER can be used to measure whether the extra cost associated with MSS+SAU is considered to constitute value for money. In other contexts, such as economic evaluations of new interventions for use in the UK's NHS, the National Institute for Health and Care Excellence (NICE[33]) uses a cost-effectiveness threshold value ($\lambda$) of £20 000–30 000 per QALY.[34] There is no agreed threshold for the perspective of costs to the UCS.

Uncertainties in the mean incremental costs and mean incremental effect (for each outcome), which determines estimated levels of cost-effectiveness, were explored using bootstrap[35] resampling: 200 replications were taken from each of the 50 imputations (in line with [27]), with resampling stratified by sex and cohort. From the replicates, cost-effectiveness planes were constructed for the base case and each scenario; each plane (one per outcome) displays the estimated mean incremental cost and mean incremental effect for each replicate,[36] with location and spread of points showing uncertainty. Additionally, cost-effectiveness acceptability curves (CEACs) were constructed for the base case and each scenario, and show the probability of MSS+SAU being cost-effective compared with SAU at a range of willingness-to-pay thresholds.[37]

In the base case (primary) economic analysis, the QALYs and costs up to T2—in line with the primary clinical-effectiveness outcome in [13]—are considered. To check the robustness of the corresponding conclusions, a number of sensitivity analyses[32] were conducted in which the assumptions of the base case analysis were varied:

1. CCA—repeats the base case, but only including participants for whom we have complete data; investigates the impact of MI
2. Per-protocol analysis—investigates how results differ if we only consider those attending at least 50% (four out of eight) of the mindfulness sessions.
3. Conduct evaluation at 12 months—extend the base case analysis to include costs and benefits up to T3, investigating whether benefits continue beyond the examination term.
4. Conduct a cost-effectiveness analysis (CEA) for CORE-OM—as for the base case, but measure benefits in terms of CORE-OM at T2 (eg, calculate the cost per CORE-OM point).
5. Conduct a CEA for WEMWBS—as for the base case, but measure benefits in terms of WEMWBS at T2 (eg, calculate the cost per WEMWBS point).

All analyses were conducted in R.[38] No costs or benefits were considered beyond 12 months, so no discounting was undertaken.[32 34]

## Patient and public involvement

Study plans were reviewed by a group comprising representatives from the UCS, the university Academic Division, student representatives and college tutors. A focus group with students who had completed mindfulness courses taught before the trial was conducted to consult about study plans prior to submission for ethics approval.

Subsequently, an advisory reference group was put together comprising student representatives, members of the UCS and other student welfare staff. This group met approximately three times per year during the trial. Study researchers attended these meetings and presented updates. The group also advised on appropriate dissemination strategies.

## RESULTS
### Participants

A total of 616 students were recruited to the trial during the academic year 2016/2017 (see figure 1): 342 were recruited during September 2016 (Michaelmas cohort), 172 to MSS+SAU and 170 to SAU; 274 were during January 2017 (Lent cohort), 137 students to each arm.

A CONSORT diagram of participants is given in the online supplemental file. Both arms (see table 1) had an approximate equal age of 23 years, but MSS+SAU had proportionately fewer females: 61% versus 65%, a nonsignificant difference. Other variables (including ethnic origin; disability; degree level; year of study and department) were well balanced across arms of the trial (see [13]).

At baseline, participants reported more psychological distress than the general UK student population (mean (total mean score)=0.76, SD=0.59),[17] but lower levels than those who attend university counselling services in the UK (total mean score=1.85, SD=0.51).[39] Our sample's mean score is just below the CORE-OM's recommended clinical cut-off score of 1 point, selected as a threshold to discriminate optimally between a clinical sample and a general population sample.[40] Participants also reported lower well-being scores than the general student population.[24]

### Resource use

The UCS has a wide range of offerings: use of the most common individual offerings (individual assessment and individual counselling sessions) and aggregates across the different types of UCS offerings between T0 and T2

**Table 1** Univariate comparisons of variables and outcomes by arm

| | | MSS+SAU (n=309) | | | SAU (n=307) | | | |
|---|---|---|---|---|---|---|---|---|
| | | n | n/Mean | SD | n | n/Mean | SD | P value |
| | Gender female (n) | 309 | 187 | | 307 | 201 | | 0.2115 |
| | Age | 306 | 23 | 5 | 306 | 23 | 6 | 0.9219 |
| Michaelmas cohort (172, 170) | Utility T0 | 172 | 0.744 | 0.114 | 168 | 0.750 | 0.121 | 0.6656 |
| | Utility T1 | 138 | 0.782 | 0.093 | 135 | 0.754 | 0.115 | 0.0257 |
| | Utility T2 | 126 | 0.786 | 0.098 | 122 | 0.735 | 0.127 | 0.0005 |
| | Utility T3 | 92 | 0.784 | 0.100 | 103 | 0.769 | 0.122 | 0.3244 |
| | QALYs T0–T2 | 116 | 0.483 | 0.045 | 109 | 0.464 | 0.058 | 0.0064 |
| | QALYs T0–T3 | 86 | 0.778 | 0.071 | 88 | 0.749 | 0.094 | 0.0237 |
| Lent cohort (137, 137) | Utility T0 | 137 | 0.767 | 0.114 | 136 | 0.782 | 0.104 | 0.2415 |
| | Utility T1 | 115 | 0.783 | 0.101 | 90 | 0.771 | 0.107 | 0.4313 |
| | Utility T2 | 109 | 0.796 | 0.111 | 94 | 0.759 | 0.112 | 0.0205 |
| | Utility T3 | 77 | 0.794 | 0.092 | 66 | 0.779 | 0.120 | 0.4044 |
| | QALYs T0–T2 | 102 | 0.316 | 0.030 | 85 | 0.312 | 0.032 | 0.3513 |
| | QALYs T0–T3 | 70 | 0.853 | 0.066 | 60 | 0.828 | 0.081 | 0.0595 |
| CORE-OM mean score | T0 | 309 | 1.0 | 0.5 | 305 | 1.0 | 0.5 | 0.4102 |
| | T1 | 255 | 0.9 | 0.5 | 227 | 1.0 | 0.5 | 0.0022 |
| | T2 | 237 | 0.9 | 0.5 | 216 | 1.1 | 0.6 | 0.0000 |
| | T3 | 169 | 0.8 | 0.5 | 169 | 0.9 | 0.6 | 0.0312 |
| WEMWBS sum score | T0 | 307 | 48.0 | 8.6 | 307 | 48.6 | 8.5 | 0.3900 |
| | T1 | 254 | 49.6 | 8.9 | 221 | 46.9 | 9.0 | 0.0009 |
| | T2 | 235 | 48.9 | 9.0 | 214 | 46.4 | 9.1 | 0.0031 |
| | T3 | 168 | 51.1 | 9.6 | 167 | 48.8 | 8.9 | 0.0243 |

Utilities and QALYs separated by cohort given differing durations to accrue QALYs.
CORE-OM, Clinical Outcomes in Routine Evaluation Outcome Measure; MSS, Mindfulness Skills for Students; QALYs, quality adjusted life years; SAU, support as usual; WEMWBS, Warwick-Edinburgh Mental Well-being Scale.

**Table 2** Aggregated UCS resource use by arm between T0 and T2

| Resource use T0 to T2 | MSS+SAU (n=309) | | SAU (n=307) | | P value |
|---|---|---|---|---|---|
| | n/Mean | SD | n/Mean | SD | |
| Used UCS services: N= | 49 | | 55 | | |
| Individual assessment sessions | 0.09 | 0.30 | 0.11 | 0.33 | 0.5937 |
| Individual counselling sessions | 0.30 | 1.31 | 0.47 | 1.70 | 0.1875 |
| Total other individual activity | 0.03 | 0.18 | 0.05 | 0.41 | 0.3637 |
| Total workshops | 0.07 | 0.33 | 0.04 | 0.25 | 0.2748 |
| Total groups | 0.03 | 0.16 | 0.02 | 0.14 | 0.5978 |

Means are the amount of each activity by student.
MSS, Mindfulness Skills for Students; SAU, support as usual; UCS, University Counselling Service.

are compared by arm in table 2. Relatively small proportions in each arm used routine UCS offerings: 16% in MSS+SAU and 18% in SAU, a non-significant difference (p=0.5199). Compared with SAU, the MSS+SAU arm used on average fewer routine UCS offerings, but none of the differences was significant (p>0.18). More detailed resource use comparisons between arms are given in the online supplemental file.

### Unit costs
Hourly costs (including employer on-costs) of employing grades 4 and 7 staff, respectively, are (see supplementary file for detailed information): £17.71 and £25.45. Grade 4 staff were administrators and support staff; grade 7 staff corresponded to UCS counsellors, or the mindfulness teacher when in relation to MSS. Unit costs for all UCS offerings are given in the online supplemental file. The two most commonly delivered UCS activities are individual assessment and counselling sessions, both delivered by grade 7 staff, with estimated unit costs per student respectively of £42.42 and £27.57. The resources and costs involved in delivering a course (for 30 students) of mindfulness are given in table 3; the resulting cost (£52.82) per student is included as a cost for all MSS+SAU students.

### Costs
Aggregate costings per student for UCS and mindfulness course use from T0 to T2 by arm are given in table 4; detailed costings by each activity are given in the supplementary file. Excluding costs of the mindfulness course, costs are lower in the MSS+SAU arm, but not significantly so (p>0.18). Overall, mean costs in the MSS+SAU arm (£66.91) are significantly (p<0.0001) higher than in the SAU (£19.42), and this is primarily driven by the costs of the mindfulness course.

### Outcomes
Unadjusted outcome comparisons are reported in table 1.

Given the different periods each cohort has to accrue QALYs, we consider utilities and QALYs separately by cohort. Utilities are plotted by arm and cohort in the online supplemental file (within each cohort/arm group, N varies by time point—see table 1). Irrespective of cohort, utilities for MSS+SAU (black) are higher (better) at T1–T3 than SAU (red); they are significantly higher at T2 (the primary outcome time point) for both cohorts (p<0.02). Compared with SAU, MSS+SAU accrued more QALYs by both T2 and T3, but the means were significantly (p=0.01 and p=0.02, respectively) higher in the

**Table 3** Resources and costs used to deliver a course of mindfulness, with capacity for 30 students

| Activities | Times per course | Duration | Staff grade | Cost (£) |
|---|---|---|---|---|
| Administrative support | 1 | 23.25 hours | 4 | 407 |
| Mindfulness teacher project management | 1 | 20.8 hours | 7 | 509 |
| Venue preparation | 8 | 60 min | 7 | 204 |
| Presession individual student contact time | 8 | 15 min | 7 | 51 |
| Group session—1st | 1 | 90 min | 7 | 38 |
| Group session—2nd–8th | 7 | 75 min | 7 | 223 |
| Postsession individual student contact time | 8 | 15 min | 7 | 51 |
| Postsession note recording | 8 | 15 min | 7 | 51 |
| Venue 'tidy-up' | 8 | 15 min | 7 | 51 |
| Total | | | | 1584 |
| Total by student (n=30) | | | | 52.82 |

**Table 4** Mean cost (per student) incurred by the UCS

| Costs T0 to T2 | MSS+SAU (n=309) | | SAU (n=307) | | |
| | n/Mean (£) | SD (£) | n/Mean (£) | SD (£) | P value |
| --- | --- | --- | --- | --- | --- |
| Individual assessment sessions | 3.98 | 12.85 | 4.56 | 14.02 | 0.5937 |
| Individual counselling sessions | 8.39 | 36.15 | 12.84 | 46.92 | 0.1875 |
| Total other individual activity | 0.60 | 5.00 | 1.24 | 11.33 | 0.3712 |
| Total workshops | 0.30 | 1.49 | 0.16 | 0.94 | 0.1781 |
| Total groups | 0.83 | 6.12 | 0.62 | 4.77 | 0.6288 |
| MSS course | 52.82 | 0.00 | 0.00 | 0.00 | |
| Total | 66.91 | 46.28 | 19.42 | 56.35 | 0.0000 |

MSS, Mindfulness Skills for Students; SAU, support as usual; UCS, University Counselling Service.

Michaelmas cohort; in the Lent cohort, MSS+SAU arm approached significantly (p=0.06) higher QALYs at T3.

At baseline (T0), CORE-OM mean scores were similar between arms, but for all subsequent points (T1–T3), MSS+SAU had significantly (p<0.03) lower (better) mean scores than SAU. Mean WEMWBS sum scores were similar at baseline (T0), but MSS+SAU scores were all significantly (p<0.02) better (higher) than SAU (T1–T3).

### Cost-effectiveness analysis

Estimates of the adjusted mean incremental costs and mean incremental outcomes (QALYs, CORE-OM improvement or WEMWBS improvement) generated from the bivariate regressions are shown in table 5, alongside the corresponding ICERs (note that CORE-OM scores have been reversed, so direction of improvement aligns with other outcomes). CEACs for the base case and scenarios S1–5 are shown in figure 2.

In the base case and all scenarios, MSS+SAU achieves significantly (95% CIs exclude zero) higher (better) outcomes, but is also significantly more expensive, than SAU. In this setting, there are no established thresholds for what ICERs would be considered cost-effective.

However, at NICE threshold of £20 000 per QALY, used in economic evaluations for the NHS, MSS+SAU would be considered cost-effective for the base case and scenarios S1–3 (where outcomes are measured in terms of QALYs).

The uncertainties surrounding cost-effectiveness conclusions are depicted in the cost-effectiveness planes in the Supplementary file. All bootstrap replicates fell in the north-east quadrant for the base case and all scenarios S1–5: this means that when evaluating uncertainty using bootstrap methods, *all* bootstrap replicates indicated the MSS+SAU was both more costly and more effective than SAU.

The probability of cost-effectiveness, illustrated by the CEACs (figure 2), rapidly moves to near certainty as the willingness-to-pay thresholds exceeds £6000 per outcome unit (base case and S1–S3: QALYs; S4: CORE-OM unit; S5: WEMWBS unit). Considering the base case and scenarios S1–S3 at the NICE thresholds noted above, there is near certainty of MSS+SAU being more cost-effective than SAU: there is very little probability of making the wrong decision about cost-effectiveness at this threshold.

**Table 5** UCS costs (£), outcomes (base case and S1–S3: QALYs; S4: CORE-OM unit; S5: WEMWBS unit) and ICER for base case and scenarios, where SAU is the reference

| Scenario | Costs (£) | | | Outcome | | | ICER |
| | Incre. dif. | 95% CI | | Incre. dif. | 95% CI | | |
| --- | --- | --- | --- | --- | --- | --- | --- |
| Base case | 48 | 40 | 56 | 0.0143 | 0.0079 | 0.0207 | 3355 |
| S1: CCA | 54 | 45 | 63 | 0.0140 | 0.0073 | 0.0208 | 3842 |
| S2: Per-protocol | 50 | 40 | 60 | 0.0160 | 0.0090 | 0.0229 | 3112 |
| S3: Up to T3 (1 year) | 47 | 35 | 58 | 0.0305 | 0.0179 | 0.0432 | 1525 |
| S4: CORE-OM T2 CE | 48 | 40 | 56 | 0.3 | 0.3 | 0.2 | 188 |
| S5: WEMWBS T2 CE | 48 | 40 | 56 | 3.0 | 1.6 | 4.5 | 16 |

Costs and outcomes adjusted for sex and cohort; additionally, outcomes adjusted by corresponding baseline outcome measurement.
CORE-OM scores reversed to align direction of improvement with other outcomes: an increase in CORE-OM here indicates an improvement (reduction) on the original scale.
CCA, complete case analysis; CORE-OM, Clinical Outcomes in Routine Evaluation Outcome Measure; ICER, incremental cost-effectiveness ratio; Incre. dif., incremental difference; QALYs, quality adjusted life years; SAU, support as usual; UCS, University Counselling Service; WEMWBS, Warwick-Edinburgh Mental Well-being Scale.

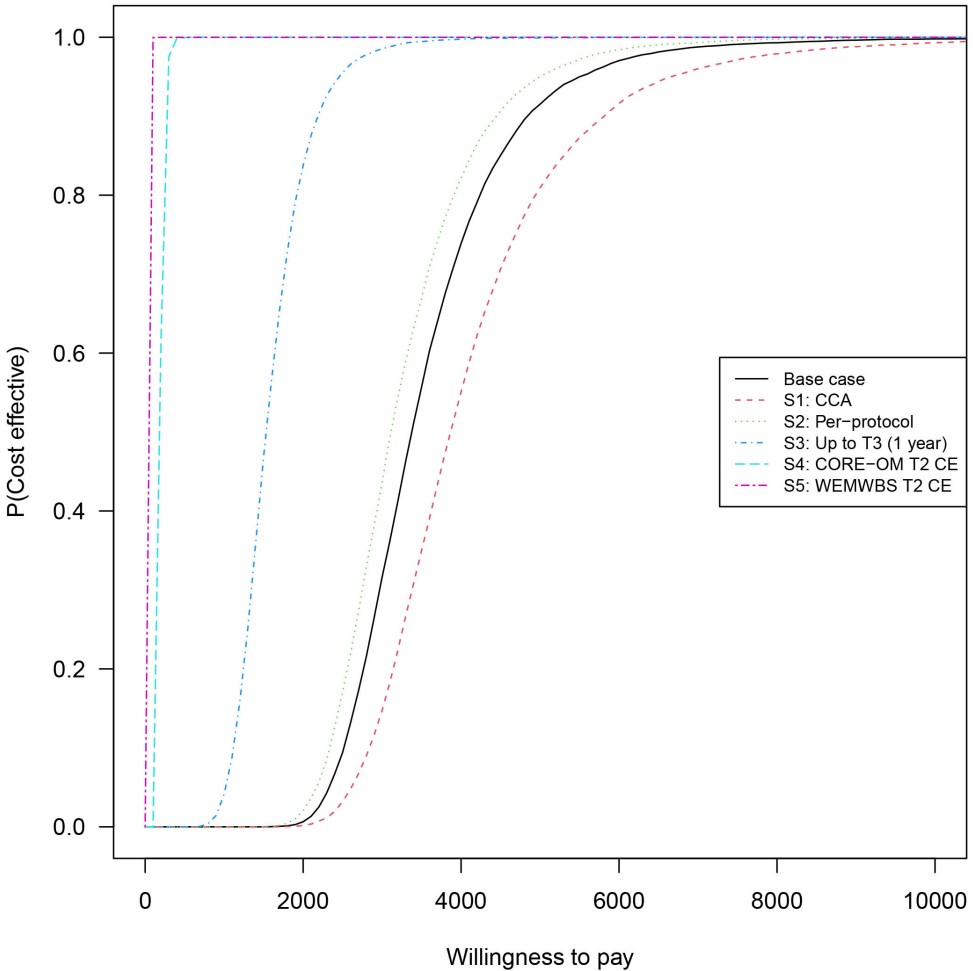

**Figure 2** Cost-effectiveness acceptability curves (CEACs) for the base-case and scenario analyses: each curve shows the probability of cost-effectiveness at a particular willingness-to-pay (λ) threshold per unit of outcome (QALYs: base case, S1-3; CORE-OM point: S4; WEMWBS point: S5). CCA, complete case analysis; CORE-OM, Clinical Outcomes in Routine Evaluation Outcome Measure; QALYs, quality adjusted life years; WEMWBS, Warwick-Edinburgh Mental Well-being Scale.

## DISCUSSION
### Main findings
There is strong evidence from here and elsewhere[13 14] that MSS leads to significantly improved outcomes. These improved outcomes come at a moderate cost: a mean additional cost of around £50 per user if mindfulness is delivered as it was here, with 30 participants per group. There are no agreed thresholds for this setting to determine whether the gains and additional costs would be considered cost-effective; however, compared with NICE thresholds (eg, £20 000/QALY), MSS+SAU would be considered cost-effective compared with SAU. The uncertainty around this decision again depends on the chosen threshold: at NICE thresholds, there is near certainty that MSS+SAU is cost-effective; also, near certainty is reached for what most would likely consider low willingness-to-pay values (see figure 2).

However, as covered below in the Study strengths and weaknesses section, this economic evaluation does not measure some of the wider costs and benefits of improving student health—such as potentially reduced use of university healthcare services (less cost to the university) and

increasing security of tuition fee income (fees not lost through student dropout). The importance of the latter point is touched on by Simpson: "As the costs and benefits of higher education become an increasingly important economic issue for students, institutions and governments, student retention will correspondingly gather increasing importance as a vital aspect of the economic analysis of higher education." (p.34, 35)[41] Additionally, though perhaps less tangibly, a stronger student mental health offering may promote student satisfaction and institutional reputation, key factors for student recruitment, in the increasingly competitive higher education setting.

### Comparison with other studies
Systematic reviews of mindfulness-based interventions for students have broadly been in favour of mindfulness improving student mental health.[42–44] Evaluations of such interventions have generally been in RCTs, but have been noted to vary in quality, typically with a high risk of bias, limited follow-up and with little consideration of safety.[42–44] In contrast, the RCT from which this economic

evaluation draws data has a lower than average risk of bias, followed students for up to 12 months and reported on safety.[13] A more recent study has also found benefits in favour of mindfulness-based interventions improving student mental health.[45]

Turning to economic evaluations of mindfulness-based interventions, we are unaware of any others in university settings. Recent systematic reviews of economic evaluations[46 47] of mindfulness-related interventions show a near exclusive focus on clinical populations, particularly for relapse prevention in major depression (but other populations are also considered, eg [48]). These reviews consider a range of interventions which incorporate mindfulness training to varying degrees; most comparable to this trial's interventions are mindfulness-based cognitive therapy (MBCT) and mindfulness-based stress reduction (MBSR). Both reviews conclude similarly for MBCT that "findings are inconclusive because of the presence of very positive results in some trials … and modest results in others" (p.143).[46] There was only one economic evaluation of an MBSR intervention,[49] with an ICER of $22 200/QALY (international dollars at 2014 prices) which the authors considered within the threshold of cost-effectiveness.

We are aware of only two articles that explore the economic impact of mindfulness outside of clinical populations.[50 51] Dongen et al[50] conducted a "cost-effectiveness analysis and return-on-investment analysis comparing mindfulness-based work site intervention to usual practice" (p.550) for governmental research institute employees. Usual practice 'dominated' the mindfulness intervention: for all outcomes considered (work engagement, general vitality and work ability), scores were statistically significant in favour of usual practice, and costs lower for this group from both employer and societal perspectives. Accordingly, the return on investment was negative. These negative findings are suggested to be due to the 'participants' low compliance with some of the intervention components' (p.555). MSS participants' intervention attendance was not dissimilar to theirs, although MSS participants appear to have been slightly more compliant with homework.[13]

Over 5 years, Klatt et al[51] compare healthcare costs (to insurers) and utilisation by university staff who received one of two interventions (one of them mindfulness) compared with matched controls: compared with controls, costs were lower for both of the interventions. There was no economic evaluation in the original trial,[52] but the interventions were suggested to be 'low-cost'[52] while unlikely to be funded by the insurer.[51]

## Study strengths and weaknesses
Contrasting with many studies in this area (see Comparison with other studies section), a key strength of this economic evaluation is that it drew on data from a large (n>600) RCT that followed up participants for up to 12 months. Additionally, to our knowledge, it is the only economic evaluation of a mindfulness intervention for students. The choice of measures used allowed QALYs to be calculated.

A key weakness of the study is the very narrow costing perspective adopted—that of the UCS. This perspective likely captures the majority of costs of directly delivering the intervention, but may well miss wider impacts. A natural perspective to adopt would be that of the university, but collecting required information for such a wide and multidomain perspective (eg, details on tuition fees and use of other student support services) would be very challenging.

A further weakness is that in costing, we have focused exclusively on the costs of staff time, as measured by salary plus 'on-costs' (eg, such as employers' pension and national insurance costs—see online supplemental file). We have not considered wider overheads or other resources (such as room hire), which will underestimate costs. In contrast to the NHS (for instance [53]), we are not aware of any pre-existing cost sources. A potentially key cost exclusion, particularly given the group based nature of the intervention, is the cost of room hire. While the UCS' operating budget does not incur a cost for room hire, there is of course a potential 'opportunity cost', by which we mean that if the rooms were not being used for MSS, they could potentially be used for something else, such as teaching. We would expect the opportunity cost to be lower where there is less 'competition' for rooms—for example, if mindfulness were delivered in the evenings, when rooms are not required for teaching.

The primary analysis was a cost-utility analysis[32] where the ICER corresponds to cost per additional QALY. Interpreting the ICER is difficult in this sector as there are no established cost-effectiveness thresholds (see Analyses section). As argued by [54], policy makers may benefit from a different analysis such as cost-benefit analysis (CBA) which 'is a comparison of interventions and their consequences in which both costs and resulting benefits (health outcomes and others) are expressed in monetary terms'.[55] Alternatives can then compared using the net monetary benefit 'which is the difference between the benefit of each treatments (expressed in monetary units) less the cost of each'.[55] However, valuing benefits in monetary terms as required by CBA is demanding, and arguably particularly so in this setting where benefits occur in many diverse domains.

## Conclusions
This economic evaluation has shown MSS+SAU accrue more benefits than SAU, but these increased benefits come at a greater cost. The main drivers of these increased costs were the costs of delivering the MSS course. Whether these benefits are considered to be worth the additional cost depends on the cost-effectiveness threshold adopted—this is not established for the university sector. Compared with the NICE threshold for the NHS, MSS+SAU would be considered highly cost-effective compared with SAU.

**Author affiliations**
[1] NIHR Applied Research Collaboration (ARC) East of England (EoE), Cambridge, UK
[2] Norwich Medical School, University of East Anglia, Norwich, UK
[3] Department of Psychiatry, University of Cambridge, Cambridge, UK
[4] Contemplative Studies Centre, Melbourne School of Psychological Sciences, Faculty of Medicine, Dentistry, and Health Sciences, University of Melbourne, Melbourne, UK
[5] Therapeutic Consultations Ltd, Cambridge, UK
[6] European Association for International Education, Amsterdam, The Netherlands
[7] Department of Kinanthropology, Charles University, Praha, Czech Republic
[8] MRC Cognition and Brain Sciences Unit, Cambridge University, Cambridge, UK

**Acknowledgements** We thank the study participants, the mindfulness teacher Elizabeth English for her development of the intervention independently of the researchers, the administrative and senior management team at the UCS, Alice Benton and Emma Howarth.

**Contributors** GD conceived the mindfulness intervention pilot on behalf of the UCS and concurrent research into the effectiveness of the provision. PBJ and GD applied for funding for the underlying randomised controlled trial. JG and PBJ produced an initial draft of the overall study protocol that was revised through discussion with all the authors. APW and GB produced an initial draft of the health economic plan which was revised through discussion with all authors. JG, GD, MV and PBJ did the study, with input from the other authors. APW led the health economic analysis, with input from JG, GB and JS. APW led drafting of this article, which was read, reviewed and approved by the other authors. PBJ is study guarantor.

**Funding** This is a summary of research funded by the University of Cambridge Vice-Chancellor's Endowment Fund, the University Counselling Service and the National Institute for Health and Care Research (NIHR) Applied Research Collaboration East of England (ARC EoE) programme. The views expressed are those of the authors and not necessarily those of the University of Cambridge, NHS, NIHR or Department of Health and Social Care.

**Competing interests** None declared.

**Patient and public involvement** Patients and/or the public were involved in the design, or conduct, or reporting, or dissemination plans of this research. Refer to the Methods section for further details.

**Patient consent for publication** Not applicable.

**Ethics approval** This study involves human participants and was approved by the Cambridge Psychology Research Ethics Committee on 25 August 2015 (reference: PRE.2015.060). Participants gave informed consent to participate in the study before taking part.

**Provenance and peer review** Not commissioned; externally peer reviewed.

**Data availability statement** Data are available upon reasonable request. Deidentified individual participant data and dictionary are available for researchers upon request from the corresponding author after approval of a proposal, with a signed data access agreement.

**ORCID iDs**
Adam P Wagner http://orcid.org/0000-0002-9101-3477
Julieta Galante http://orcid.org/0000-0002-4108-5341
Maris Vainre http://orcid.org/0000-0001-9570-3726
Peter B Jones http://orcid.org/0000-0002-0387-880X

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
