## [Reviewer comments · BMJ Open]

ARTICLE DETAILS

TITLE (PROVISIONAL)	Cost-effectiveness of providing university students with a mindfulness-based intervention to reduce psychological distress: economic evaluation of a pragmatic randomised controlled trial
AUTHORS	Wagner, Adam; Galante, Julieta; Dufour, Géraldine; Barton, Garry; Stochl, Jan; Vainre, Maris; Jones, Peter

VERSION 1 – REVIEW

REVIEWER	Tiemens, Bea Radboud Universiteit Faculteit der Sociale Wetenschappen, Behavioural Science Institute
REVIEW RETURNED	30-May-2023

GENERAL COMMENTS	This could be a very interesting and relevant study, as students in many countries experience a lot of stress during the study period. This is a nicely done cost-effectiveness study, but I have serious reservations about the purpose of the study. I have some major points and some smaller concerns. Major points: 1. The focus/goal of the study is not clear to me. The title and abstract talk about "increase resilience to stress", but resilience is not measured. The outcome measures are symptoms on the CORE-OM and quality of life with the WEMWBS. Moreover, the first paragraph of the Background talks about the need for "preventive intervention". But the study does not measure whether anything is prevented (e.g. depressive disorder or study dropout), it seems not a prevention study. So my question is, is it an intervention study or a prevention study? And depending on that, an intervention for what or prevention of what?2. As a consequence of my previous point, it is not clear to me what group of students is involved. It just says that students were recruited into two cohorts, but how and where? Were these random students or students seeking help? And if they did not seek help what is 'support as usual' and for what? Results section 3.2. states that only a small proportion of students in each arm used SAU also in the SAU arm. So did the rest of the students in the SAU arm receive nothing? And how severe were the problems of this population? It is good that the baseline scores on the two instruments were similar in both conditions, but were these scores high or low?3. Even though the CORE-6D is a preference-based tool in the UK to measure mental health-specific health state utility, internationally it is less common for calculating QALYs than the EQ5D. Please provide some more information on the calculation of the utility scores. Other points:
--

	4. The abstract lacks follow-up duration. It is not clear what the measurement moment of the results is. 5. The timing of the second follow-up measurement, T2, is not clear, page 3: "The date of exam term follow-up, T2, was common across cohorts". Does this mean that that measurement moment was the same for both cohorts, but thus the period between T0 and T2 differed? 6. Page 8, 3.3. Unit costs: what is employing grade 4 and 7, what kind of professionals? 7. It is unfortunate that in the discussion section, the authors do not consider the significance for the university. In the conclusion, they ask whether "these benefits are considered to be worth the additional cost.... – this is not established for the university sector". Surely they could have made a preamble here? E.g. what is the cost of a student dropping out after the first or second year? Or the cost of students who delay one or two years? But again, it is actually not clear what to reflect on, because in my opinion the focus of the study is not clear, see my first point.
--	--

REVIEWER	Fjorback, Lone Aarhus University Hospital, Denmark, Aarhus University
REVIEW RETURNED	11-Jun-2023

GENERAL COMMENTS	Review This is an impressive, comprehensive, well conducted, well researched and well written paper on the economic evaluation of Mindfulness Skills for Students. The overall finding on exact cost related to teach MSS + support as usual (SAU) for university students when compared to SAU supports the hypothesis that MSS is cost-effective according to the NICE guidelines. In current times with depression being the leading cause of disability this paper could be improved by including research on the consequences and severity of the mental health crisis in youth, the direct and indirect cost related to stress, anxiety and depression and the preventive potential for teaching students mental health skills in training the mind. 4 papers on these issues could be included in background and discussion. MBSR for students Even through mindfulness are offered as prevention the students seeking MBSR may have symptoms within a clinical range and still the treatment and prevention potential is great https://www.frontiersin.org/articles/10.3389/fpsyg.2021.722771/full Training the mind MBSR prevention Juul L et. al Altered self-reported resting state mediates the effects of Mindfulness-based stress reduction on mental health. A longitudinal path model analysis within a community-based randomized trial with 6-months follow-up Frontiers accepted for publication Training the mind MBCT treatment https://www.sciencedirect.com/science/article/pii/S0006322322014354 Economic evaluation alongside RCT found that 5 and 10 years before patients with chronic stress disorders received treatment they experienced more unemployment and more sickness benefit. MBSR reduced risk of receiving disability pension when compared to enhanced treatment as usual. Also, health care cost were reduced. https://mindfulness.au.dk/fileadmin/_migrated/content_uploads/Mindfulness_therapy__economic_analysis_.pdf
--

VERSION 1 – AUTHOR RESPONSE

Reviewer: 1

Prof. Bea Tiemens, Radboud Universiteit Faculteit der Sociale Wetenschappen, Pro Persona Mental Health Care

Comments to the Author:

This could be a very interesting and relevant study, as students in many countries experience a lot of stress during the study period. This is a nicely done cost-effectiveness study, but I have serious reservations about the purpose of the study.

I have some major points and some smaller concerns.

Major points:

1. The focus/goal of the study is not clear to me. The title and abstract talk about "increase resilience to stress", but resilience is not measured. The outcome measures are symptoms on the CORE-OM and quality of life with the WEMWBS. Moreover, the first paragraph of the Background talks about the need for "preventive intervention". But the study does not measure whether anything is prevented (e.g. depressive disorder or study dropout), it seems not a prevention study. So my question is, is it an intervention study or a prevention study? And depending on that, an intervention for what or prevention of what?

Reply: *We understand the points made by this reviewer. Our main trial publication [1] defines the use of the term "resilience to stress" in the context of the trial: "Our primary hypothesis was that provision of mindfulness courses would reduce students' psychological distress during the examination period, when stress peaks, compared with support as usual. A reduction in distress while under a universal stressor (examinations) was deemed an indicator of resilience to stress" [p.e73, 1]. However, we are happy to refer more directly to our primary outcome in this manuscript, so we have replaced mentions of resilience with mentions of psychological stress reduction in the two places where this phrase was used, namely title and abstract. We believe this has made our manuscript clearer.*

We do, however, believe that the intervention we have tested can be classified as a preventative intervention. Psychological or mental distress is a concept encompassing a range of disturbing or unpleasant mental or emotional experiences, in which symptoms of depression and anxiety are central. It is not a mental health condition in itself, but there is extensive evidence that it constitutes a transdiagnostic marker of increased risk of mental, and even physical, health conditions [2-4]. We think, therefore, that any intervention that reduces psychological distress, by reducing the risk of health conditions is effectively preventing ill-health. We clarify this in the introduction.

2. As a consequence of my previous point, it is not clear to me what group of students is involved. It just says that students were recruited into two cohorts, but how and where? Were these random students or students seeking help? And if they did not seek help what is 'support as usual' and for what? Results section 3.2. states that only a small proportion of students in each arm used SAU also in the SAU arm. So did the rest of the students in the SAU arm receive nothing? And how severe were the problems of this population? It is good that the baseline scores on the two instruments were similar in both conditions, but were these scores high or low?

Reply: *We have refined and added further information to the Participant section of the Methods: "The trial inclusion criteria were: (a) undergraduate or postgraduate UoC students; (b) who were interested in attending at least seven sessions of the course. The exclusion criteria were: (a) severe anxiety or depression at that time; (b) severe mental illness such as psychosis; (c) recent*

bereavement; and (d) any other serious health problem that would affect their ability to engage with the course. The selection criteria were assessed by the students themselves.

The study was advertised widely to the student community, using physical (posters) social media (eg Facebook and Twitter) and information sessions – see [5] for further detail. Where students agreed to take part, they were emailed a personal link to an online baseline questionnaire. Upon completing this questionnaire, participants underwent 1:1 randomisation to receive either: an eight week mindfulness course adapted for university students plus support as usual (MSS+SAU); or SAU alone. SAU consisted of the possibility of accessing, if the student desired, comprehensive UCS support in addition to other health support available from the UoC and its colleges, and from health services including the National Health Service, external to the UoC. Participants randomised to SAU were guaranteed a space in the following year's mindfulness courses and were requested to inform the team if they decided to learn mindfulness elsewhere during the follow-up period. Further detail on the trial can be found in [5] and [1].”

We believe this covers the Reviewer's initial points relating to how the cohorts were identified, recruited, and what they could access as SAU. Additionally, we have added more information to the beginning of the Results section to contextualise the scores of our cohort:

“At baseline, participants reported more psychological distress than the general UK student population (mean (total mean score)=0.76, SD=0.59) [6], but lower levels than those who attend university counselling services in the UK (total mean score =1.85, SD=0.51) [7]. Our sample's mean

score is just below the CORE-OM's recommended clinical cut-off score of 1 point, selected as a

threshold to discriminate optimally between a clinical sample and a general population sample [8]. Participants also reported lower wellbeing scores than the general student population [9].”

3. Even though the CORE-6D is a preference-based tool in the UK to measure mental health-specific health state utility, internationally it is less common for calculating QALYs than the EQ5D. Please provide some more information on the calculation of the utility scores.

Reply: *The pre-specified health economic analysis plan (HEAP) – already referenced in the paper – includes an Appendix which lists the items from the CORE-OM and other detail used in the calculation*

of the CORE-6D utility. The HEAP is available online [10] – we have added additional text to the

methods to highlight this:

“Scores generated from the survey are converted to an index score, that can be used as a valuation of all health states represented by the CORE-6D: details of the steps for these calculations are provided in an Appendix to the HEAP [10].”

We repeat the relevant text from the HEAP at the end of this response to reviewers for ease of reference.

Other points:

4. The abstract lacks follow-up duration. It is not clear what the measurement moment of the results is.

Reply: *We have added this detail to the abstract to address the Reviewer's point:*

“Primary follow-up duration was 5 and 7 months for the two recruitment cohorts.”

[The differing timings are explored/explained further in the following point]

5. The timing of the second follow-up measurement, T2, is not clear, page 3: "The date of exam term follow-up, T2, was common across cohorts". Does this mean that that measurement moment was the same for both cohorts, but thus the period between T0 and T2 differed?

Reply: Yes, the Reviewer is correct: the 'measurement moment' at T2 was the same across cohorts, but the duration between T0 and T2 differed by cohort – this is addressed in this paragraph from the methods:

"Students were recruited in two cohorts: at the beginning of the autumn term (Michaelmas) term and at the beginning of the spring (Lent) term. Both cohorts were followed-up at: point of recruitment (T0); after the delivery of the mindfulness courses (SAU having *not* received the course) (T1); during the exam term (T2); and 12 months after recruitment (T3). The date of exam term follow-up, T2, was common across cohorts, but the date of other follow-up points differed by cohort: the relationships are depicted in Figure 1."

However, we have added the following text to the end of this paragraph to further clarify this point.

"Thus, the follow-up duration at T2 was approximately 7 months for the Michaelmas cohort and 5 months for the later recruited Lent cohort."

Our analysis adjusts for this differing follow-up through use of a covariate for cohort, and is also the reason for the separately reporting QALYs for each cohort in Table 1 (as noted in the Table caption: "Utilities and QALYs separated by cohort given differing durations to accrue QALYs"). This is inline with the analysis used in the main outcomes papers [1, 11].

6. Page 8, 3.3. Unit costs: what is employing grade 4 and 7, what kind of professionals?

Reply: We have added the following text to the Unit Costs section of the Results to clarify:

"Grade 4 staff were administrators and support staff; grade 7 staff corresponded to UCS counsellors, or the mindfulness teacher when in relation to MSS."

7. It is unfortunate that in the discussion section, the authors do not consider the significance for the university. In the conclusion, they ask whether "these benefits are considered to be worth the additional cost... – this is not established for the university sector". Surely they could have made a preamble here? E.g. what is the cost of a student dropping out after the first or second year? Or the cost of students who delay one or two years? But again, it is actually not clear what to reflect on, because in my opinion the focus of the study is not clear, see my first point.

Reply: We have added the following text to the Discussion section:

"However, as covered below in the `Study strengths and weaknesses`, this economic evaluation does not measure some of the wider costs and benefits of improving student health – such as potentially reduced use of university healthcare services (less cost to the university) and increasing security of tuition fee income (fees not lost through student drop-out). The importance of the latter point is touched on by Simpson: "As the costs and benefits of higher education become an increasingly important economic issue for students, institutions and governments, student retention will correspondingly gather increasing importance as a vital aspect of the economic analysis of higher education" [p.34-35, 12]. Additionally, though perhaps less tangibly, a stronger student mental health offering, may promote student satisfaction and institutional reputation, key factors for student recruitment, in the increasingly competitive higher education setting."

Reviewer: 2

Dr. Lone Fjorback, Aarhus University Hospital, Denmark Comments to the Author:

Review

This is an impressive, comprehensive, well conducted, well researched and well written paper on the economic evaluation of Mindfulness Skills for Students. The overall finding on exact cost related to teach MSS + support as usual (SAU) for university students when compared to SAU supports the hypothesis that MSS is cost-effective according to the NICE guidelines.

In current times with depression being the leading cause of disability this paper could be improved by including research on the consequences and severity of the mental health crisis in youth, the direct and indirect cost related to stress, anxiety and depression and the preventive potential for teaching students mental health skills in training the mind.

4 papers on these issues could be included in background and discussion.

MBSR for students

Even though mindfulness are offered as prevention the students seeking MBSR may have symptoms within a clinical range and still the treatment and prevention potential is great

<https://www.frontiersin.org/articles/10.3389/fpsyg.2021.722771/full>

Training the mind MBSR prevention

Juul L et. al Altered self-reported resting state mediates the effects of Mindfulness-based stress reduction on mental health. A longitudinal path model analysis within a community-based randomized trial with 6-months follow-up Frontiers accepted for publication

Training the mind MBCT treatment

<https://www.sciencedirect.com/science/article/pii/S0006322322014354>

Economic evaluation alongside RCT found that 5 and 10 years before patients with chronic stress disorders received treatment they experienced more unemployment and more sickness benefit. MBSR reduced risk of receiving disability pension when compared to enhanced treatment as usual. Also, health care cost were reduced.

https://mindfulness.au.dk/fileadmin/migrated/content/uploads/Mindfulness_therapy_economic_analysis_.pdf

Reply: Thank you for the feedback and suggested references. This manuscript is focused on the economic evaluation as mental health aims and outcomes have been addressed extensively in previous trial publications [1, 11]. However, we have included those publications we think best contribute to the narrative. We have also added some additional context on the mental health crisis in the background section.

To address Reviewer 1's point 3, material extracted from the Health Economic Analysis Plan (HEAP) [10], available at: [https://www.anzctr.org.au/Steps11and12/369222-\(Uploaded-23-08-2019-20-0235\)-Study-related document.pdf](https://www.anzctr.org.au/Steps11and12/369222-(Uploaded-23-08-2019-20-0235)-Study-related+document.pdf)

15 Appendix 1: Calculating CORE-6D utility values from CORE-OM items

The details given here are based on Mavranouzouli, Brazier [13] and Mavranouzouli [14].

The utility score is based on CORE-OM items 1, 8, 15, 16, 21 and 33. These items are recoded as in Table A1 to form the CORE-6D (CO6D) items (*this table assumes that scores for item 21 have already been reversed*). The total score of the emotional component of the CORE-6D is given by the sumscore based on CORE-6D items 1, 15, 16, 21 and 33. The corresponding utility for each health state can then be determined from Table A2 using the emotional component score and responses to the physical item (recoded CORE-OM item 8).

Missing items on any of the underlying CORE-OM items (1, 8, 15, 16, 21 and 33) lead to a missing utility value.

Table A1: Conversion for taking CORE-OM items and forming CORE-6D items.

CORE-6D component	Emotional component					Physical item
CORE-6D (CO6D) item	CO6D-1	CO6D-15	CO6D-16	CO6D-21	CO6D-33	CO6D-8
CORE-OM original item	1	15	16	21	33	8
0	0	0	0	0	0	0
1	1	1	1	0	1	1
CORE-OM original 2 response levels	1	1	1	1	1	1
3	2	2	2	1	2	2
4	2	2	2	2	2	2

Table A2: Utility values for the CORE-6D health states indexed by the total value of the emotional component and CO-6D-8 values. Draws heavily on Table 8, p.391, Mavranouzouli, Brazier [13].

CORE-6D sum score of emotional component	Response level of CO6D-8		
	0	1	2
0	0.95	0.92	0.81
1	0.94	0.90	0.80
2	0.87	0.84	0.73
3	0.80	0.77	0.66
4	0.72	0.69	0.58
5	0.64	0.61	0.50
6	0.55	0.52	0.41
7	0.47	0.43	0.32
8	0.38	0.35	0.24
9	0.30	0.26	0.16
10	0.24	0.20	0.10

References

- Galante, J., et al., *A mindfulness-based intervention to increase resilience to stress in university students (the Mindful Student Study): a pragmatic randomised controlled trial*. The Lancet Public Health, 2017.
- Russ, T.C., et al., *Association between psychological distress and mortality: individual participant pooled analysis of 10 prospective cohort studies*. BMJ : British Medical Journal, 2012. **345**: p. e4933.
- Dalgleish, T., et al., *Transdiagnostic approaches to mental health problems: Current status and future directions*. J Consult Clin Psychol, 2020. **88**(3): p. 179-195.
- Ela, P., et al., *How do the prevalence and relative risk of non-suicidal self-injury and suicidal thoughts vary across the population distribution of common mental distress (the p factor)? Observational analyses replicated in two independent UK cohorts of young people*. BMJ Open, 2020. **10**(5): p. e032494.

5. Galante, J., et al. *Protocol for the Mindful Student Study: a randomised controlled trial of the provision of a mindfulness intervention to support university students' well-being and resilience to stress*. *BMJ Open*, 2016. **6**, DOI: 10.1136/bmjopen-2016-012300.
6. Core System Group. *CORE system user manual*. 2015 15 September 2015]; Available from: <http://www.coreims.co.uk/index.html>.
7. Stiles, W.B., M. Barkham, and S. Wheeler, *Duration of psychological therapy: relation to recovery and improvement rates in UK routine practice. [corrected]*. *Br J Psychiatry*, 2015. **207**(2): p. 115-22.
8. Connell, J., et al., *Distribution of CORE-OM scores in a general population, clinical cut-off points and comparison with the CIS-R*. *Br J Psychiatry*, 2007. **190**: p. 69-74.
9. Stewart-Brown, S. and K. Janmohamed, *Warwick-Edinburgh mental well-being scale user guide*, University of Warwick, Editor. 2008: Coventry.
10. Wagner, A.P. and G. Barton. *Mindfulness student study (MSS): Health economic analysis plan (HEAP)*. 2018 [cited 2018; Available from: [https://www.anzctr.org.au/Steps11and12/369222\(Uploaded-23-08-2019-20-02-35\)-Study-related](https://www.anzctr.org.au/Steps11and12/369222(Uploaded-23-08-2019-20-02-35)-Study-related) document.pdf.
11. Galante, J., et al., *Effectiveness of providing university students with a mindfulness-based intervention to increase resilience to stress: 1-year follow-up of a pragmatic randomised controlled trial*. *Journal of Epidemiology and Community Health*, 2021. **75**(2): p. 151.
12. Simpson, O., *The costs and benefits of student retention for students, institutions and governments*. *Studies in Learning, Evaluation Innovation and Development*, 2005. **2**(3): p. 3443.
13. Mavranzouli, I., et al., *Estimating a Preference-Based Index from the Clinical Outcomes in Routine Evaluation–Outcome Measure (CORE-OM): Valuation of CORE-6D*. *Medical Decision Making*, 2013. **33**(3): p. 381-395.
14. Mavranzouli, I., *The derivation of a preference-based measure for people with common mental health problems from the Clinical Outcomes in Routine Evaluation Outcome Measure (CORE-OM)*, in *Health Economics and Decision Science, School of Health and Related Research (SchARR)*. 2014, University of Sheffield: Sheffield.

VERSION 2 – REVIEW

REVIEWER	Tiemens, Bea Radboud Universiteit Faculteit der Sociale Wetenschappen, Behavioural Science Institute
REVIEW RETURNED	13-Oct-2023
GENERAL COMMENTS	All points were clearly addressed, I have no further comment.